# Predictive Values of Blood-Based RNA Signatures for the Gemcitabine Response in Advanced Pancreatic Cancer

**DOI:** 10.3390/cancers12113204

**Published:** 2020-10-30

**Authors:** David Piquemal, Florian Noguier, Fabien Pierrat, Roman Bruno, Jerome Cros

**Affiliations:** 1Acobiom, 1682 Rue de la Valsière, 34790 Grabels, France; noguier@acobiom.com (F.N.); pierrat@acobiom.com (F.P.); bruno@acobiom.com (R.B.); 2Department of Pathology, Beaujon Hospital-Université de Paris–INSERM U1149, 92110 Clichy, France; jerome.cros@aphp.fr

**Keywords:** pancreatic cancer, predictive diagnosis, liquid biopsy, gemcitabine

## Abstract

**Simple Summary:**

Pancreatic adenocarcinoma is predicted to be the 2nd cause of death by cancer in Western Countries in 2023. Most patients are diagnosed at an advanced stage for which chemotherapy is the main treatment. In this study we developed through an innovative approach, a simple nine genes blood RNA-based signature that predicts sensitivity to gemcitabine, one of the main regimens in combination with nab-paclitaxel or alone in less fit patients.

**Abstract:**

Pancreatic ductal adenocarcinoma (PDAC) is expected to be the second cause of cancer death by 2022. For nearly 80% of patients, diagnosis occurs at an advanced, nonsurgical stage, making such patients incurable. Gemcitabine is still an important component in PDAC treatment and is most often used as a backbone to test new targeted therapies and there is, to date, no routine biomarker to predict its efficacy. Samples from a phase III randomized trial were used to develop through a large approach based on blood-based liquid biopsy, transcriptome profiling, and machine learning, a nine gene predictive signature for gemcitabine sensitivity. Patients with a positive test (41.6%) had a significantly longer progression free survival (PFS) (3.8 months vs. 1.9 months *p* = 0.03) and a longer overall survival (OS) (14.5 months vs. 5.1, *p* < 0.0001). In multivariate analyses, this signature was independently associated with PFS (HR = 0.5 (0.28–0.9) *p* = 0.025) and OS (HR = 0.39 (0.21–0.7) *p* = 0.002).

## 1. Introduction

Pancreatic ductal adenocarcinoma (PDAC) is currently the fourth leading cause of cancer death in Western Countries and if no action is taken, it will rise to be second in 2022. Despite recent advances in chemotherapy regimens, the survival rates of PDAC remain very low, around 7% at 5 years, while survival now reaches 87% for breast cancer [1,2]. This is mainly due to a late diagnosis, mostly at the metastatic stage, and the important chemoresistance of the tumor. While the therapeutic options have slightly broadened, there is no efficient predictive biomarker to stratify patients [3]. PDAC treatment is based on few regimens of chemotherapies: Folfirinox (median OS = 11.1 months [4]) and gemcitabine/nab-paclitaxel (median OS = 8.5 months [5]) or, for unfit patients, a single drug regimen with gemcitabine or 5-fluorouracil (5-FU) (median overall survival; OS = 6.8 months [4]). Gemcitabine is, therefore, still an important component in PDAC treatment and is most often used as a backbone to test new targeted therapies. There is, to date, no routine biomarker to predict gemcitabine efficacy. Human equilibrative nucleoside transporter 1 (hENT1), encoded by the *SLC29A1* gene, the transmembrane gemcitabine transporter, appeared to be the most promising predictive biomarker for gemcitabine sensitivity, but the only immunohistochemistry antibody with predictive properties is not commercially available [6,7]. Other genes involved in the gemcitabine metabolism have also been proposed such as *dCK*, *RRM1/2*, and *CDA* but these studies did not result in routine diagnostic tests [8]. Furthermore, the microenvironment plays a major role in drug resistance. For gemcitabine for instance, it was shown in a mouse model that tumor associated fibroblasts can uptake and metabolize a significant part of the gemcitabine [8]. Similarly, it was recently shown that mast cells caused resistance to gemcitabine/nabpaclitaxel by reducing apoptosis following activation of the TGF-β signaling [9]. While these biological features of the tumor are critical to drug sensitivity/resistance, their assessment by tissue-based tests (i) requires biopsies or surgical specimens and (ii) ignores spatial/temporal tumor heterogeneity, as multiple sampling is classically not performed. The failure of the CO-101 biomarker-based clinical trial to meet its endpoint because of ineffective stratification demonstrates the importance of these two issues [10].

To date, blood-based liquid biopsy has not become part of the routine workup in PDAC, but its potential applications are rapidly growing. RNA- or DNA-based liquid biopsy is less invasive and could turn conventional research into therapeutically actionable molecular alterations. In addition, it was shown to have a prognostic value and allowed the early prediction of relapse [11]. However, today, circulating tumor DNA (ctDNA) can only be detected in half of patients, mostly because the sensitivity of the technique is still too low. In contrast, whole blood RNA-based signatures that capture tumor-induced changes in the circulating transcriptome may prove useful, especially in tumor types with a low level of ctDNA [12].

The goal of this study was to develop a blood RNA-based signature to predict gemcitabine sensitivity in advanced PDAC. Blood samples collected during a prospective multicenter randomized double-blind placebo-controlled phase III study that evaluated the efficacy and safety of Masitinib in combination with gemcitabine in patients with advanced/metastatic PDAC were used (ClinicalTrials.gov Identifier: NCT00789633) [13]. A summary of the clinical and biological data is presented in Table 1 and Table 2.

## 2. Results

Gene expression analyses were established on peripheral blood cell samples collected before the start of therapy (naive patients). Using a next-generation sequencing-based transcriptomic analysis, we selected 62 differentially expressed genes. It is worth noting that genes described to participate in gemcitabine metabolism in tumors such as *SCL29A1* (*Solute Carrier Family 29 Member 1-hent1*), *CDA* (*Cytidine Deaminase*), or *dCK* (*Deoxycytidine kinase*) were expressed but not differentially in this comparative study. Then, the validation of these putative biomarkers was transferred to a standard and affordable quantitative real-time PCR workflow; finally, an RNA signature was established.

Quantitative real-time PCR analyses were performed on 62 genes of interest and two housekeeping genes. Statistical modelling was done on the Delta.Cp (DCp) values according to the method described by Livak and Schmittgen [14]. The cox model-based selection provided two separate RNA-blood signature based on nine distinct genes found to be significantly associated with prognosis and they presented interesting molecular features. In these two signatures (OS and PFS), six genes are associated with the OS: *ABCC1* (*ATP Binding Cassette Subfamily C Member 1*), *ARL4C* (*ADP Ribosylation Factor Like GTPase 4C*), *LYN* (*LYN Proto-Oncogene, Src Family Tyrosine Kinase*), *NME4* (*NME/NM23 Nucleoside Diphosphate Kinase 4*), *PPIB* (*Peptidylprolyl Isomerase B*), *UBE2H* (*Ubiquitin Conjugating Enzyme E2 H*). On the other hand, five genes are associated with the PFS: *ARL4C*, *NME4*, *ALDOA* (*Aldolase, Fructose-Bisphosphate A*), *GAB3* (*GRB2 Associated Binding Protein 3*), and transporters like *SLC35E2B* (*Solute Carrier Family 35 Member E2B*). This gene set composed of nine unique genes was used to build two GE Scores: one for OS prediction and one for PFS prediction.

The predictive value of the two blood-based RNA signatures allowed patient stratification in two groups whose characteristics at baseline were fairly comparable between those with a positive test (GE+) and those with a negative test (GE-) except for a higher proportion of patients with a performance status at 1 and slightly more advanced tumors in the GE-OS group (Appendix A). Patients with a positive PFS test (GE + PFS) (41.6%) had a significantly longer progression free survival ((3.8 months (95% CI = 3.5–7.9 months) vs. 1.9 months (95% CI = 1.8–3.8 months), HR = 1.8 (95% CI = 1–3.1), *p* = 0.03)). Patients with a positive OS test (GE + OS) had a significant longer overall survival (14.5 months (95% CI = 10.6–19.6 months; 1-year survival rate = 65%) vs. 5.1 months (95% CI = 4–7.4 months; 1-year survival rate = 12%), HR = 3.2 (95% CI = 1.8–5.7), *p* < 0.0001) (Figure 1). The specificity and the sensibility of the model were 74% and 81%, respectively, to predict overall survival under gemcitabine treatment.

At last, contingency testing of GE score with clinical and tumor characteristics (Table 1 and Table 2) did not reveal any statistically significant associations. In multivariate analyses, the GE + OS and the GE + PFS were independently associated with a prolonged OS and PFS, respectively, in a statistically significant manner.

## 3. Discussion

While this study is exploratory with a reduced number of patients, it used clinical data and samples from a large multicenter registered trial to ensure good quality, homogeneity, and minimal bias. As this study included only advanced patients naive of treatment, the classical pathological variables that have a very strong impact such as tumor differentiation, the N stage, the R0 status were not available and therefore not included. Yet, the most important clinical variables for advanced patients (metastatic status, performance status, albumin level, etc.) were included in the multivariate analyses to ensure that the value of the GE score was independent of them. This study paves the way for the use of blood-based RNA signatures that currently remain uncommon in PDAC. Sakai et al. reported a diagnostic test with a similar approach, but this is the first report of a blood-based predictive signature [15]. It will require to be validated on an external cohort and on patients that did not receive gemcitabine to assess whether the GE score only has a predictive value or also carry a prognostic value, potentially broadening its use to select patients fit for surgery for instance. The whole process has been transferred toward a standard and affordable quantitative real-time PCR workflow that could be easily integrated in daily practice as it only requires a 2.5 mL blood sample, that can be part of a standard medical consultation. A tool that selects gemcitabine sensitive PDAC will be important in clinical practice as it was demonstrated that gemcitabine was as efficient as 5-FU in combination with nab-paclitaxel in advanced lesions. Similarly, in unfit patients for FOLFIRINOX, gemcitabine and 5-FU are two valid options. Selecting good responders to gemcitabine could improve patient care and this study demonstrate the feasibility of RNA-based blood test. Adapting this approach to other therapy could lead to an integrated test to make personalized medicine a reality in PDAC.

Interestingly, the biomarkers expressed in the whole blood seem to mirror some of the known metabolic pathways in solid tumors, strengthening our belief in their use as liquid biopsy biomarkers. Gemcitabine is a pro-drug that requires cellular uptake and serial phosphorylation to become pharmacologically active. One mechanism responsible for gemcitabine resistance is dysregulation of the proteins participating in gemcitabine metabolism pathways, including deficiency of the hENTs/Solute Carrier Family, downregulation of the rate-limiting enzyme dCK, and upregulation of RRM1/RRM2 or CDA [16,17]. While these classical “tumor cells” markers of gemcitabine sensitivity were not part of the GE signature, one of the genes *ABCB1*, an efflux pump of the (ABC) transporter family proteins was demonstrated to be involved in gemcitabine resistance [18]. In addition, blood mRNA that were selected in this signature may or may not be directly derived from the tumor and there is no way in this setting on human samples to assess their origin. They may very well also be the “blood transcriptomic” consequence of a particular combination of tumor and stromal cells that are sensitive or resistant to gemcitabine. Finally, there may be an important bias in the tumor mRNA that are released in the blood based on their size, their stability, but also the associated protein, leading to discrepancies between the tumor and the blood level.

## 4. Materials and Methods

### 4.1. Patients and Sample Collection

Blood samples were collected in PAXgene Blood RNA tube (PreAnalytiX, Hombrechtikon, Switzerland) from a prospective, multicenter, randomized, double-blind, two-parallel group, placebo-controlled phase III trial evaluating the safety and efficacy of masitinib plus gemcitabine against placebo plus gemcitabine in chemotherapy-naive PDAC patients (ClinicalTrials.gov Identifier: NCT00789633). Gemcitabine (1000 mg/m^2^) was administered according to standard clinical practice. Treatments were administered until progression, intolerance, or patient withdrawal, with disease progression assessed via CT scan according to the RECIST criteria every 8 weeks. In the event of a treatment-related grade 3 or 4 adverse event (AE), treatment interruption or blinded dose reduction was permitted according to the predefined criteria. The investigation was carried out in accordance with the Declaration of Helsinki. The 60 patients, all from the gemcitabine arm (Clinical Trial NCT00789633) were included for this study. A summary of clinical and biological data is presented in Appendix A. Blood samples were collected prior to the initiation of gemcitabine treatment.

### 4.2. Gene Expression Analysis via qPCR

The total RNA from blood samples was extracted using a PAXgene Blood RNA Kit V2 (QIAGEN, Hilden, Germany) on an automated QIAcube according to the manufacturer’s protocols. The RNA purity and quantity were controlled using a NanoDrop ND-1000 spectrophotometer, and the RNA integrity was controlled with an Agilent 2200 TapeStation. The following quality requirements were applied: RNA concentration >30 ng/µL, RNA absorbance (260/280 nm) > 1.8, and RINe > 6.5. The gene expression analyses were performed using a LightCycler^®^ 480 SYBR Green I Master in a 10 µL final reaction volume according to the manufacturer’s protocol using a LightCycler^®^ 480 System II Instrument (Roche Diagnostics, Meylan, France). PCR Primers and assay were validated with the procedure described in Appendix A and Section 4.

### 4.3. Statistical Analysis: Selection of Candidate Genes

Based on the DCp values, candidate genes were selected to test their significance using the R software (R.3.1.2 64 bits), Bioconductor package v2.14, glmnet package v1.9-8, and maxstat package v0.7-22. We performed a Cox-net regression using the glmnet package for gene selections. Our methodology is based on the Elastic-Net method, first introduced by Zou and Hastie [19]. It is an extension of the penalized methods of Lasso [20] and Ridge regression [21].

### 4.4. Statistical Analysis: Gene Expression-Based (GE) Score

Our data set was randomized with a training set made from 3/4 of the randomized data sets, and 1/4 left for the testing set. First, we selected iteratively a combination of genes (one signature) between 5 and 15 items. For each gene, a β coefficient was computed from fitting the Cox-net regression. The products of the β coefficient and the DCp values for each patient and for each gene selected in the signature were summed to obtain a single value labelled as the index score. A mobile cut off around the median value of the index score was calculated in order to maximize the *p*-value for a Log rank test between the divided training set. Then the index score was calculated on the testing set and the *p*-value from a Log rank test was stored. These steps were repeated 60,000 times. Then, the better model was selected based on two criteria: firstly, based on the lowest *p*-value achievable by the log rank test on the testing set; secondly, to prevent the selection of a model that would represent random variation and yield the best fit we developed a ranking system. In this ranking system each gene was ranked by calculating a score based on the frequency of appearance of each gene across the 60,000 signatures computed. Using this score each signature is weighted by the sum of all its genes score generating a signature ranking. This ranking had to be in the top 0.01% of all the computed signature to be vetted by the algorithm.

### 4.5. Statistical Analysis: Univariate and Multivariate Analysis

Prediction from the GE score OS and the GE score PFS were added to the clinical covariates from the Table 1 and Table 2 and performed both univariate and multivariate analysis tested using Cox proportional hazards. The *p*-value results are in Table 1 and Table 2 and Appendix A for univariates analyses and multivariate analyses.

## 5. Conclusions

In this retrospective study we demonstrated on a prospectively collected biological collection the validity of a rapid, cost-efficient, blood RNA-based test to predict gemcitabine sensitivity in advanced PDAC patients. Today, nearly 80% of cancer patients do not have genetic profiling available at the initial oncology consultation [22], and virtually none if we consider gene expression information. Combining this test with germline testing for alterations in the homologous recombination genes could represent the first step of precision medicine in PDAC care.

## Figures and Tables

**Figure 1 cancers-12-03204-f001:**
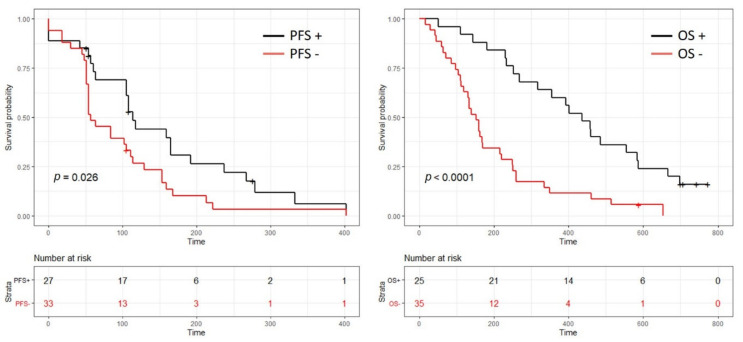
Kaplan–Meier plots for both progression free survival (PFS) (left side) and overall survival (OS) (right side). The black curve identifies the patient whose blood-based transcriptomic profile is favorable to gemcitabine treatment as a 1st line therapy. The red curve identifies the patient whose blood-based transcriptomic profile is unfavorable (a low response and risk of potential toxicities) to gemcitabine treatment as a 1st line therapy.

**Table 1 cancers-12-03204-t001:** Multiple regression analysis of overall survival factors by multivariable analysis.

Analyze	Univariate		Multivariate	
Result	HR (95% CI for HR)	*p*-Value	HR (95% CI for HR)	*p*-Value
GE score OS + prediction	0.31 (0.17–0.56)	0.000095	0.39 (0.21–0.7)	0.002
CA 19–9 (U/mL)	1 (1–1)	0.28		
Albumin (g/L)	0.97 (0.94–1)	0.16		
QLQ-C30	1 (1–1)	0.0027	1.02 (1.0–1.04)	0.015
Body mass index	0.98 (0.92–1)	0.5		
ECOG PS	1.8 (0.96–3.2)	0.067		
Monocyte count (per µL)	2 (0.9–4.6)	0.09		
Tumor localization				
Head	0.86 (0.51–1.5)	0.59		
Body	1.1 (0.62–1.9)	0.81		
Tail	1.5 (0.85–2.7)	0.16		
Clinical stage	0.34 (0.17–0.68)	0.0024	0.41 (0.2–0.83)	0.014

Abbreviations: QLQ-C30; European Organization for Research and Treatment of Cancer (EORTC) Quality of Life Questionnaire Core 30 item global health status. GE; gene expression. CA19–9; carbohydrate antigen 19–9. ECOG PS; Eastern Cooperative Oncology Group Performance Status.

**Table 2 cancers-12-03204-t002:** Multiple regression analysis of progression free survival factors by multivariable analysis.

Analyze	Univariate		Multivariate	
Result	HR (95% CI for HR)	*p*-Value	HR (95% CI for HR)	*p*-Value
GE score PFS + prediction	0.55 (0.32–0.95)	0.032	0.5 (0.28–0.9)	0.025
CA 19–9 (U/mL)	1 (1–1)	0.47		
Albumin (g/L)	1 (0.96–1)	0.81		
QLQ-C30	1 (1–1)	0.038	1.02 (1.0–1.04)	0.026
Body mass index	0.99 (0.93–1.1)	0.76		
ECOG PS	2 (1–3.8)	0.045	1.6 (0.8–3.1)	0.17
Monocyte count (per µL)	0.73 (0.29–1.8)	0.49		
Tumor localization				
head	1.2 (0.69–2.1)	0.52		
body	1.1 (0.64–1.9)	0.7		
tail	1.5 (0.84–2.8)	0.16		
Clinical classification	0.7 (0.35–1.4)	0.32		

Abbreviations: QLQ-C30; European Organization for Research and Treatment of Cancer (EORTC) Quality of Life Questionnaire Core 30 item global health status. GE; gene expression. CA19–9; carbohydrate antigen 19–9. ECOG PS; Eastern Cooperative Oncology Group Performance Status.

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
