# Peer review of "Predictive Values of Blood-Based RNA Signatures for the Gemcitabine Response in Advanced Pancreatic Cancer"

_cancers, 2020, doi:10.3390/cancers12113204_

Round 1

Reviewer 1 Report

Piquemal and Cols present a very interesting paper looking for a predictive RNA-signature using blood from the control arm of the Masitinib trial. Overall the question they are trying to answer is relevant to the field although some questions arise reading the MS.

Major questions

  1. in the results, first-second paragraph, some of the info given could be added to M&M and maybe a flowchart could help to better follow the rational to select the first 62 genes and how they end up in 9 genes. 
  2. It would be relevant to see how those 9 genes do in terms of OS an PFS by themselves. Are the 9 genes equally relevant for OS/PFS
  3. Is not clear why there are different number of GE+/- in the OS group and in the PFS group. If they are the same patients, shouldn't we have the same number of GE+ patients for PFS and for OS?
  4. The authors state that there is a slight difference in the amount of advanced patients in GE- vs GE+ groups but there as a clear statistical difference in the table given. In addition, could it be that the gene-signature is related to the stage of the patients?
  5. The authors should use a different cohort than the one used to get the signature to test for OS/PFS (discovery and validation cohorts are needed in this type of approache). In addition, could the authors take out the non metastatic patients to see if the signature is really predictive of OS/PFS in the metastatic setting? Additional cohort also needed. 
  6. There is no explanation in the Table to explain the meaning of * and **
  7. The conclusion is not sustained by data shown. The cohort has both locally advanced and metastatic patients and although the trial was prospective is not clear if this was a pre-planned analysis or not. In addition this might more like a prospective-retrospective study than a proper prospective study.

Minor changes

  1. Line 16 trough TYPO
  2. Line 41 a reference is missing
  3. Lines 86-88, too long and unclear. please, rephrase.
  4. Statistical part of M&M should be reorganised so that it is easier to follow.

Author Response

Dear Editor, dear Reviewer,

Thank you very much for the interest in our work and your constructive criticisms to improve the manuscript. Please find below a point by point response. Changes are highlighted in the text by the MS Word “change tracking function”.

Comments and Suggestions for Authors

Piquemal and Cols present a very interesting paper looking for a predictive RNA-signature using blood from the control arm of the Masitinib trial. Overall the question they are trying to answer is relevant to the field although some questions arise reading the MS.

Major questions

  1. in the results, first-second paragraph, some of the info given could be added to M&M and maybe a flowchart could help to better follow the rational to select the first 62 genes and how they end up in 9 genes. 

Indeed, and we have replaced the technical info in M&M section and reformulated first-second paragraph in order to better explain the process ending up to 9 genes as biomarkers.

  1. It would be relevant to see how those 9 genes do in terms of OS an PFS by themselves. Are the 9 genes equally relevant for OS/PFS

We agree with the reviewer: the 9 genes are not equally relevant for OS/PFS. We added more explanations regarding the 2 distinct signatures: one for OS prediction and one for PFS prediction, in the results, lines 80-86.

  1. Is not clear why there are different number of GE+/- in the OS group and in the PFS group. If they are the same patients, shouldn't we have the same number of GE+ patients for PFS and for OS?

There is a difference because there is two distinct signatures clustering the patients in distinct groups depending on the outcome (PFS or OS). Rare patients may have a long OS and a poor PFS with the first line of treatment. Thus, a patient may be GE+OS and GE-PFS and vice-versa.

  1. The authors state that there is a slight difference in the amount of advanced patients in GE- vs GE+ groups but there as a clear statistical difference in the table given. In addition, could it be that the gene-signature is related to the stage of the patients?

It is true that the distribution of metastatic patient is different between the GE+OS and GE-OS but not between the GE+PFS and GE-PFS. Without a cohort of patient with no treatment or a non gemcitabine-based treatment we cannot formally ruled out that the GE test also has a prognostic value. We added this limit in the discussion.

  1. The authors should use a different cohort than the one used to get the signature to test for OS/PFS (discovery and validation cohorts are needed in this type of approach). In addition, could the authors take out the non-metastatic patients to see if the signature is really predictive of OS/PFS in the metastatic setting? Additional cohort also needed. 

We agree with the reviewer that it would be interesting to test the signature in the metastatic and non-metastatic setting but unfortunately the number of patients is not sufficient, especially for a proper multivariate analysis. Clearly this is a preliminary study and the GE score will require to be externally validated on several cohorts and we hope this report will help us secure the access to these additional cohort. Yet, this study used samples collected during a phase three randomized trial ensuring a good quality of the clinical data and of the homogeneity of the biological collection. This is highlighted lines 128-130.

  1. There is no explanation in the Table to explain the meaning of * and **

We discarded * ; p-value in itself is sufficient.

  1. The conclusion is not sustained by data shown. The cohort has both locally advanced and metastatic patients and although the trial was prospective is not clear if this was a pre-planned analysis or not. In addition this might more like a prospective-retrospective study than a proper prospective study.

The collection of the blood samples was performed prospectively during the phase three trial with the planned objective to develop a companion diagnostic for the trial promoter (AB Science Company (Paris, France)). For this purpose, we conducted a transcriptomic and genomic study on the masitinib+gemcitabine arm. In a second time, following internal observations, we explored the gemcitabine arm in a non-planned analysis. We agree with the reviewer that this is indeed a retrospective analysis on a prospective cohort. We have changed the conclusion to add the retrospective nature of the study (line 210).

Minor changes

  1. Line 16 trough TYPO: The correction is done
  2. Line 41 a reference is missing: The reference is added
  3. Lines 86-88, too long and unclear. please, rephrase. We have looked carefully your proposal, and after appropriate consideration, we discarded these lines because they are a limited input in this communication.
  4. Statistical part of M&M should be reorganised so that it is easier to follow. We agreed and simplified the thought process in order to better understand the methodology without giving all the algorithmic details.

Reviewer 2 Report

To the authors

David Piquemal and colleagues uncovered a 9 gene predictive signature in Pancreatic ductal adenocarcinoma (PDAC) to be predictive for gemcitabine sensitivity by using a liquid biopsy, transcriptome profiling, and machine learning. The manuscript covers an interesting topic well, nonetheless, there are few sections that deserve to be restructured, in order to achieve the level and comprehensive overview that a journal like Cancers would aim to.

Major points to consider in subsequent versions:

1.Methods: the authors employed qPCR. They state that SYBR Green have been used. Because SYBR Green I dye binds to any double-stranded DNA—including nonspecific double-stranded DNA sequences—it may generate false-positive signals. Moreover, primer optimization is sometimes necessary to improve the performance of SYBR Green assays. Furthermore, Multiplex PCR cannot be done when using SYBR Green. Finally, a dissociation or “melt” of the PCR products is highly recommended for SYBR Green assays, which lengthens the protocol, and requires visual analysis of the peaks.  Conversely, TaqMan allows specific hybridization between probe and target, required to generate a fluorescent signal, significantly reducing background and false positives. Two or more specific targets may be detected in the same reaction when the probes are labeled with different dyes. Additionally, multiplex PCR can reduce cost and improve precision. Post-PCR processing is eliminated, saving time. The authors should comment on this and provide a clear short statement about the strengths and weakness of this study.

2.Clinical outcome analysis. The authors analyzed differentially clinical outcome impacted by transcriptomic profiles that is favourable to gem treatment. This is fine, as long as the authors point out how the stratified the patients (over the median vs. below the median and/or quartile, and, more generally, class boundary). Moreover, despite the authors provide a multivariate analysis (with slightly statistical significant p values and weak HR for QLQ-30),  as a prognostic marker in terms of both overall and progression-free survival, the clinical characteristics of those patients can deeply impact the HR (i.e. nodal status, TNM, grade and histological subtype, R0 or R1 if surgery was performed, neo- or adjuvant therapy received, etc.). We acknowledge that this can be beyond the scope of the manuscript and anyway not possible with a retrospective in silico interrogation. Nonetheless, if co-variates suitable for multivariate statistical analyses are not available this should be mentioned as a study limitation, especially if hazards proportionality is not respected and COX multivariate model cannot be performed. Indeed, those are important information and should be provided in order to propose the elaborated score as potentially valuable in a clinical setting. Otherwise, this analysis can only be hypothesis-generating and the association with survival should be tuned down while highlighting the lack of validation to date (validation cohort available? The author discussed this point in line 117, but the impact of their results should be tuned down accordingly)

General comments: An advice to broader thrill broad oncology filed involved readership might be to add few key points while introducing and discussing the hypothesis inferred by their findings, regarding gemcitabine sensitivity/refractoriness in light of previously (recently) reported findings. As an example, in the introduction, the authors briefly quote a seminal manuscript (ref. 7) while summarizing about gemcitabine transporter. In the referenced manuscript immune evasion, PDAC carcinogenesis and invasiveness (angiogenesis and metastatic potential) are also discussed. This section can be slightly expanded, comprising novel findings of this topic and the insights about tumour immune-microenvironment, leading to the emergence of aberrant signalling pathways as critical factors modulating central gene-expression signatures that fuel pancreatic tumour both directly and indirectly, by shaping the immune milieu (i.e. via WNT/CTNNB1 in nodal invasive PDAC). Moreover, the authors state that “tool that select gemcitabine sensitive PDAC will be important in clinical practice as it was demonstrated that gemcitabine was as efficient as 5-FU in combination with nab-paclitaxel in advanced lesions. Similarly, in unfit patients for FOLFIRINOX, gemcitabine and 5-FU are two valid options”. I personally miss some important translational aspect potentially related to this aspect, pointing towards a potential Achilles’ heel of PDAC that might be exploited therapeutically in the future. Indeed, tumour-stroma interactions are of key importance for PDAC progression. Cancer-associated fibroblasts (CAFs) and mast cells (MC) affected the sensitivity of PDAC cells to gemcitabine/nab-paclitaxel. The MCs have been uncovered to drive resistance to drugs by reducing apoptosis, by activating the TGF-β signalling, (i.e.PMID: 30866547).

Author Response

Dear Editor, dear Reviewer,

Thank you very much for the interest in our work and your constructive criticisms to improve the manuscript. Please find below a point by point response. Changes are highlighted in the text by the MS Word “change tracking function”.

Major points to consider in subsequent versions:

  1. 1.Methods: the authors employed qPCR. They state that SYBR Green have been used. Because SYBR Green I dye binds to any double-stranded DNA—including nonspecific double-stranded DNA sequences—it may generate false-positive signals. Moreover, primer optimization is sometimes necessary to improve the performance of SYBR Green assays. Furthermore, Multiplex PCR cannot be done when using SYBR Green. Finally, a dissociation or “melt” of the PCR products is highly recommended for SYBR Green assays, which lengthens the protocol, and requires visual analysis of the peaks.  Conversely, TaqMan allows specific hybridization between probe and target, required to generate a fluorescent signal, significantly reducing background and false positives. Two or more specific targets may be detected in the same reaction when the probes are labeled with different dyes. Additionally, multiplex PCR can reduce cost and improve precision. Post-PCR processing is eliminated, saving time. The authors should comment on this and provide a clear short statement about the strengths and weakness of this study.

The assay was developed to meet to the MIQE guidelines [Bustin SA et al, 2009] and, Clinical and Health Authorities’ requirements. We established a complete analytical validation of the real-time PCR assay. This validation required a strict primer selection process and the following aspects and specifications have been addressed: 

  • Primers selection using Primer3Plus web tool (bioinformatics.nl/cgi-bin/primer3plus/primer3plus.cgi/) [Untergasser A, Nijeen H et al, 2007],

  • In silico analysis of PCR primers:
    • First analysis of primers specificity using the UCSC Genome Bioinformatics Group of the University of California, Santa Cruz (In Silico PCR; http://genome.ucsc.edu/index.html). This study is completed using Primer-blast tool on The National Center for Biotechnology Information NCBI (ncbi.nlm.nih.gov),
    • Second analysis of primers specificity focused on interactions (hairpin, self-dimer or hetero-dimer) using the “OligoAnalyzer 3.1” web tool (http://eu.idtdna.com/analyzer/Applications/OligoAnalyzer/) to avoid important PCR equilibrium destabilizations,
    • Analysis of primers efficiency: The secondary structure of single stranded nucleic acids can be predicted using the Mfold software (http://mfold.rna.albany.edu/?q=mfold/download-mfold). By determining the secondary structure of single-stranded nucleic acids, it is possible to identify regions that are easily accessible by the primers to initiate DNA elongation.

Following this in silico study, we initiated in vitro analysis.

  • In vitro analysis of PCR primers
    • First test of specificity via PCR product sizing, controlled using gel-based sizing by electrophoresis on agarose gel technique,
    • Second test of specificity via PCR product sequencing, controlled via the Sanger-based sequencing method.
    • Third test of specificity via negative controls, consisted in controlling possible cross reactivity of primers during the PCR process. For each gene, two negative controls have been used to highlight any cross reactivity of primers such as the formation of homo- or heterodimers, PCR contamination, genomic DNA contamination, etc:
  • A "NTC" control (no template control) which corresponds to a sample run in the exact same manner as the other Real-Time PCR reactions, but in which the cDNA matrix/template has been voluntary omitted. This control allows detecting any external contamination or other factors that could result in a non- specific increase in the fluorescence signal.
  • A "No RT" control (no reverse transcriptase control) which corresponds to a sample run in the exact same manner as the other Real-Time PCR reactions, but in which the cDNA matrix has been voluntary replaced by a RNA matrix.
    • Fourth test of specificity via control of amplicon Tm, consisted in controlling the melting temperature/ dissociation temperature of the double-stranded DNA PCR product amplified. The Tm must be unique for each gene,
    • Test of efficiency: The efficiency of the RT-PCR amplification for each PCR reactions ranges between 88% and 103%, demonstrating that the PCR efficiencies are approved (Thomas D Schmittgen and Kenneth J Livak, Nature Protocols 2008; http://miqe.gene-quantification.info, 2013).

Based on these controlled PCR parameters, we established analytical performance of the assay with definition of the limits of blank (LoB), the limits of detection (LoD) and the limits of quantification (LoQ), as well intra- and inter-patient variabilities.

We concluded that given results obtained, both the repeatability and reproducibility of the Real-Time PCR assay have been confirmed. The analytical performance of the assay has thus been validated.

Thus, we established a standard operating procedure (SOP) with 7 acceptance criteria, before the predictive value analysis of the signature:

  1. The LoB of NoRT controls must be validated,
  2. The LoB of NTC controls must be validated,
  3. All genes have to be measured (genes of interest GOI and housekeeping genes HKG),
  4. Measured Cp value of each gene (GOI and HKG) is replicated 5 times and the measurement of a gene is accepted if at least 4 values are measured,
  5. Repeatability of measures: Measured Cp value of each gene (GOI and HKG) is repeated 5 times and the measurement of a gene is accepted if at least 4 values are gathered to less than 0.2 Cp,
  6. Limit of Quantification (LoQ): Measured Cp value of each gene (GOI and HKG) must be inferior to defined LoQ,
  7. Temperature of melting (Tm): Measured Tm value of each gene (GOI and HKG) must be unique and closed to defined Tm values, with a maximal standard deviation of 0.5°C.

In order to add this information without complicating the M&M section, we have copied this in a supplemental M&M section (lines 178-179).

2.Clinical outcome analysis. The authors analyzed differentially clinical outcome impacted by transcriptomic profiles that is favourable to gem treatment. This is fine, as long as the authors point out how to stratified the patients (over the median vs. below the median and/or quartile, and, more generally, class boundary). Moreover, despite the authors provide a multivariate analysis (with slightly statistical significant p values and weak HR for QLQ-30),  as a prognostic marker in terms of both overall and progression-free survival, the clinical characteristics of those patients can deeply impact the HR (i.e. nodal status, TNM, grade and histological subtype, R0 or R1 if surgery was performed, neo- or adjuvant therapy received etc.). We acknowledge that this can be beyond the scope of the manuscript and anyway not possible with a retrospective in silico interrogation. Nonetheless, if co-variates suitable for multivariate statistical analyses are not available this should be mentioned as a study limitation, especially if hazards proportionality is not respected and COX multivariate model cannot be performed. Indeed, those are important information and should be provided in order to propose the elaborated score as potentially valuable in a clinical setting. Otherwise, this analysis can only be hypothesis-generating and the association with survival should be tuned down while highlighting the lack of validation to date (validation cohort available? The author discussed this point in line 117, but the impact of their results should be tuned down accordingly).

We agree with the reviewer that key clinical parameter such the pTNM, nodal status have a very strong prognostic value and may undermine the value of prognostic/predictive biomarkers if they are omitted from the analyses. These clinical parameters are of course mandatory in studies that deals with patients whose tumor could be resected. In our study, all patients were locally advanced/metastatic at diagnosis and naïve of treatment. These parameters were therefore not available but we did include the in the analyses the metastatic status together with the other classical clinical parameters for this type of advanced patients (PS/ albumin, etc.). We added this information in the discussion and the limits of the absence of a validation cohort is pointed out (lines121-124).

General comments: An advice to broader thrill broad oncology filed involved readership might be to add few key points while introducing and discussing the hypothesis inferred by their findings, regarding gemcitabine sensitivity/refractoriness in light of previously (recently) reported findings. As an example, in the introduction, the authors briefly quote a seminal manuscript (ref. 7) while summarizing about gemcitabine transporter. In the referenced manuscript immune evasion, PDAC carcinogenesis and invasiveness (angiogenesis and metastatic potential) are also discussed. This section can be slightly expanded, comprising novel findings of this topic and the insights about tumour immune-microenvironment, leading to the emergence of aberrant signalling pathways as critical factors modulating central gene-expression signatures that fuel pancreatic tumour both directly and indirectly, by shaping the immune milieu (i.e. via WNT/CTNNB1 in nodal invasive PDAC). Moreover, the authors state that “tool that select gemcitabine sensitive PDAC will be important in clinical practice as it was demonstrated that gemcitabine was as efficient as 5-FU in combination with nab-paclitaxel in advanced lesions. Similarly, in unfit patients for FOLFIRINOX, gemcitabine and 5-FU are two valid options”. I personally miss some important translational aspect potentially related to this aspect, pointing towards a potential Achilles’ heel of PDAC that might be exploited therapeutically in the future. Indeed, tumour-stroma interactions are of key importance for PDAC progression. Cancer-associated fibroblasts (CAFs) and mast cells (MC) affected the sensitivity of PDAC cells to gemcitabine/nab-paclitaxel. The MCs have been uncovered to drive resistance to drugs by reducing apoptosis, by activating the TGF-β signalling, (i.e.PMID: 30866547).

We completely agree with the reviewer that in addition to the tumor cell own sensibility to a drug, there is also a major (if not superior) effect of the stroma cells, CAF and immune. Here the signature was derived from circulating mRNA that may or may not originate from the tumor, meaning that they may not fully relate to the main local resistance pathways. In addition, there might be a selection in tumor-derived mRNA that eventually reach the blood (size, stability, associated proteins etc…). We therefore feel that it is difficult to completely mirror the meaning of the circulating mRNA with the tumor and its microenvironment. For instance, if one could predict the PDAC subtype from blood-derived mRNA, the most predictive mRNA may not be part of the tumor subtype original signature but rather reflect how the subtype affect the circulating transcriptome.

We have expanded the introduction on the role of the stroma (lines 45-52) and have added the comments above in the discussion section (lines 150-152).

Round 2

Reviewer 1 Report

I have reviewed the answers and I think that the points below have not been addressed. The authors have answered but I do not think the answers are enough to accept the paper. Below I add the more relevant points and in red some comments.   A) Is not clear why there are different number of GE+/- in the OS group and in the PFS group. If they are the same patients, shouldn't we have the same number of GE+ patients for PFS and for OS?

There is a difference because there is two distinct signatures clustering the patients in distinct groups depending on the outcome (PFS or OS). Rare patients may have a long OS and a poor PFS with the first line of treatment. Thus, a patient may be GE+OS and GE-PFS and vice-versa. Still do not understand. If there are two different signatures the title of the paper is misleading.

B)The authors should use a different cohort than the one used to get the signature to test for OS/PFS (discovery and validation cohorts are needed in this type of approach). In addition, could the authors take out the non-metastatic patients to see if the signature is really predictive of OS/PFS in the metastatic setting? Additional cohort also needed.   

We agree with the reviewer that it would be interesting to test the signature in the metastatic and non-metastatic setting but unfortunately the number of patients is not sufficient, especially for a proper multivariate analysis. Clearly this is a preliminary study and the GE score will require to be externally validated on several cohorts and we hope this report will help us secure the access to these additional cohort. Yet, this study used samples collected during a phase three randomized trial ensuring a good quality of the clinical data and of the homogeneity of the biological collection. This is highlighted lines 128-130. I am not doubting the quality but the methodology and the answer is not adequate although a statistician should be asked.

C) The conclusion is not sustained by data shown. The cohort has both locally advanced and metastatic patients and although the trial was prospective is not clear if this was a pre-planned analysis or not. In addition this might more like a prospective-retrospective study than a proper prospective study.  

The collection of the blood samples was performed prospectively during the phase three trial with the planned objective to develop a companion diagnostic for the trial promoter (AB Science Company (Paris, France)). For this purpose, we conducted a transcriptomic and genomic study on the masitinib+gemcitabine arm. In a second time, following internal observations, we explored the gemcitabine arm in a non-planned analysis. We agree with the reviewer that this is indeed a retrospective analysis on a prospective cohort. We have changed the conclusion to add the retrospective nature of the study (line 210). Again is a methodological issue that takes into account the two previous points

Author Response

Response to Reviewer 1 Comments

Point 1: I have reviewed the answers and I think that the points below have not been addressed. The authors have answered but I do not think the answers are enough to accept the paper. Below I add the more relevant points and in red some comments.   A) Is not clear why there are different number of GE+/- in the OS group and in the PFS group. If they are the same patients, shouldn't we have the same number of GE+ patients for PFS and for OS?

There is a difference because there is two distinct signatures clustering the patients in distinct groups depending on the outcome (PFS or OS). Rare patients may have a long OS and a poor PFS with the first line of treatment. Thus, a patient may be GE+OS and GE-PFS and vice-versa. Still do not understand. If there are two different signatures the title of the paper is misleading.

Response 1:

We agree with the reviewer and we modify the title “Predictive values of blood-based RNA signatures for the gemcitabine response in advanced pancreatic cancer”.

Point 2: B)The authors should use a different cohort than the one used to get the signature to test for OS/PFS (discovery and validation cohorts are needed in this type of approach). In addition, could the authors take out the non-metastatic patients to see if the signature is really predictive of OS/PFS in the metastatic setting? Additional cohort also needed.   

We agree with the reviewer that it would be interesting to test the signature in the metastatic and non-metastatic setting but unfortunately the number of patients is not sufficient, especially for a proper multivariate analysis. Clearly this is a preliminary study and the GE score will require to be externally validated on several cohorts and we hope this report will help us secure the access to these additional cohort. Yet, this study used samples collected during a phase three randomized trial ensuring a good quality of the clinical data and of the homogeneity of the biological collection. This is highlighted lines 128-130. I am not doubting the quality but the methodology and the answer is not adequate although a statistician should be asked.

Response 2:

We agree that we have not a standard study design using discovery & validation cohorts, more appropriate from a statistical point of view. But the study was done retrospectively and thus we did not have planned to include only metastatic patients (45 patients out of 60). We used the data available of the whole cohort (locally advanced and metastatic), able to predict for all PDAC patients at a non-surgical stage. As indicated in Materials and Methods (Statistical analysis: Gene Expression-based (GE) Score, §4.4), we used this cohort to simulate multiple training and test sample to build a robust model that can be used on a validation cohort and this point is our next step.

Point 3: C) The conclusion is not sustained by data shown. The cohort has both locally advanced and metastatic patients and although the trial was prospective is not clear if this was a pre-planned analysis or not. In addition this might more like a prospective-retrospective study than a proper prospective study.  

The collection of the blood samples was performed prospectively during the phase three trial with the planned objective to develop a companion diagnostic for the trial promoter (AB Science Company (Paris, France)). For this purpose, we conducted a transcriptomic and genomic study on the masitinib+gemcitabine arm. In a second time, following internal observations, we explored the gemcitabine arm in a non-planned analysis. We agree with the reviewer that this is indeed a retrospective analysis on a prospective cohort. We have changed the conclusion to add the retrospective nature of the study (line 210). Again is a methodological issue that takes into account the two previous points

Response 3:

We agreed and we repeatedly clear any concern regarding the used methodology: the cohort investigated was not recruited with controlled clinical parameters defined by our investigation because our study is retrospective. We used data available to us to answer a non-pre-planned analysis. But we used an innovative workflow to test and find two signatures that enabled us to have a significant prediction on derivated traini

Reviewer 2 Report

The authors have clarified several of the questions I raised in my previous review. Most of the major problems have been addressed by this revision.

No further comment from this reviewer.

Author Response

Dear Colleague,

Thank you for your time and considertion.

Best regards

David Piquemal